# Metabolomics Reveals the Impact of Overexpression of Cytosolic Fructose-1,6-Bisphosphatase on Photosynthesis and Growth in *Nannochloropsis gaditana*

**DOI:** 10.3390/ijms25126800

**Published:** 2024-06-20

**Authors:** Zhengying Zhang, Yanyan Li, Shuting Wen, Shu Yang, Hongmei Zhu, Hantao Zhou

**Affiliations:** 1State Key Laboratory of Marine Environmental Science, Xiamen University, Xiamen 361000, China; zhangzhengying@stu.xmu.edu.cn (Z.Z.); yanyanli2016@stu.xmu.edu.cn (Y.L.); shutingwen@163.com (S.W.); yangshummm@163.com (S.Y.); 2College of Ocean and Earth Sciences, Xiamen University, Xiamen 361000, China; flyzhu324@163.com; 3State-Province Joint Engineering Laboratory of Marine Bioproducts and Technology, Xiamen University, Xiamen 361000, China

**Keywords:** *Nannochloropsis gaditana*, *cyFBPase*, biomass, photosynthesis, metabolic

## Abstract

*Nannochloropsis gaditana*, a microalga known for its photosynthetic efficiency, serves as a cell factory, producing valuable biomolecules such as proteins, lipids, and pigments. These components make it an ideal candidate for biofuel production and pharmaceutical applications. In this study, we genetically engineered *N. gaditana* to overexpress the enzyme fructose-1,6-bisphosphatase (cyFBPase) using the Hsp promoter, aiming to enhance sugar metabolism and biomass accumulation. The modified algal strain, termed NgFBP, exhibited a 1.34-fold increase in cyFBPase activity under photoautotrophic conditions. This modification led to a doubling of biomass production and an increase in eicosapentaenoic acid (EPA) content in fatty acids to 20.78–23.08%. Additionally, the genetic alteration activated the pathways related to glycine, protoporphyrin, thioglucosides, pantothenic acid, CoA, and glycerophospholipids. This shift in carbon allocation towards chloroplast development significantly enhanced photosynthesis and growth. The outcomes of this study not only improve our understanding of photosynthesis and carbon allocation in *N. gaditana* but also suggest new biotechnological methods to optimize biomass yield and compound production in microalgae.

## 1. Introduction

Microalgae are photosynthetic microorganisms that use light energy, water, and carbon dioxide to produce algae biomass. Photosynthetic microorganisms such as microalgae are used in a wide range of fields [1]. In contrast to plants, microalgae can fix CO_2_ during photosynthesis more efficiently [2,3]. Microalgal biomass has emerged as a viable alternative for biofuel production [4], finding applications in fuel, feed, nutraceuticals, and pharmaceuticals [5]. However, research conducted in the field has been restricted to a few microalgal model species, with most microalgal diversity still unexplored and even fewer studies investigating the physiological responses of different species under similar culture conditions. In the past decade, genetic modification tools have offered a new insight into genetic transgenesis [6,7,8]. The engineering and optimization of algal strains and metabolic pathways provide a powerful tool and efficient strategy to increase biomass and lipid productivity, with the promise of advancing the research and development of economically feasible algae-to-biofuel technology [9,10].

*Nannochloropsis* species have a rich history of production and application, serving as a vital source of microalgae for aquaculture bait and eicosapentaenoic acid (EPA) production [11,12]. All species within the genus *Nannochloropsis* are specialized autotrophs, with most members extensively studied and established as models for the molecular physiology and genetic engineering of microalgae [13,14]. The genome size of *Nannochloropsis* species is approximately 30 Mb, with seven identified species, some of which have been sequenced [15]. Photosynthetic efficiency serves as a significant limitation to biomass accumulation. Improving photosynthesis to enhance the conversion of light energy into biomass is a crucial objective in genetic enhancement efforts for microalgae [16]. Previous research has explored various strategies to enhance photosynthetic efficiency, including managing spectral distribution and light intensity during microalgae cultivation to optimize external conditions [17]. Furthermore, endeavors have been made to refine conditions affecting photosynthetic efficiency through immobilization techniques, optimal management of algal cell suspension culture density, and genetic modifications of algal photosynthetic systems [18,19].

The research on photosynthesis in *Nannochloropsis* species encompasses various aspects, including adaptation to high- and low-light conditions, analysis of the photosynthetic system’s structure, and the molecular genetic enhancement of photosynthesis genes. In a specific study, a mutant strain (hlr1) of *N. oceanica* exhibited robust tolerance to high light intensity, attributed to reduced reactive oxygen species (ROS) production stemming from alterations in Photosystem I (PSI). This reduced oxidative damage and promoted growth under high-light conditions [20]. These findings underscore the potential of investigating photosynthesis in *Nannochloropsis* species to deepen our understanding of its regulatory mechanisms and identify targets for genetic enhancement, ultimately enhancing biomass production.

Microalgae are fast-growing photoautotrophic organisms that can be cultivated to exploit sunlight energy to transform atmospheric CO_2_ into organic biomass [21]. Microalgal biomass is a sustainable and renewable feedstock for value-added products such as animal feeds and foods, health supplements [22], pharmaceuticals, personal care products, biofuels, and chemical feed stocks [23]. Fructose-1,6-bisphosphatase (FBPase1; EC 3.1.3.11) is the main regulatory step of sucrose biosynthesis. Fructose-1,6-bisphosphatase catalyzes the breakdown of fructose-1,6-bisphosphate to fructose-6-phosphate (F6P) and Pi. Two types of fructose-1,6-bisphosphatase have been described in microalgae [24]: the cytosolic enzyme, which is involved in sucrose synthesis and gluconeogenesis [25], and chloroplastidial isoforms. CpFBPase [26] is involved in the pathway of RuBP recontribution in the Calvin–Benson cycle. Cytosolic FBPase [27,28] is one of the rate-limiting enzymes involved in sucrose biosynthesis and catalyzes the first reaction in the conversion of triose phosphates to sucrose. Most of the data indicate that carbon partitioning is significantly reduced in sucrose biosynthesis under the cytosolic FBPase loss-of-function mutants [29]. The overexpression of FBPase from cyanobacteria in tobacco showed increased photosynthetic CO_2_ fixation as well as increased biomass. In *Chlamydomonas reinhardtii*, the overexpression of chloroplast FBPase had no significant accelerating effect on growth rate or biomass [30]. The overexpression of cytosolic FBPase in *Brassica napus* also significantly promoted the growth and biomass of tobacco [31]. These findings suggest that cytosolic FBPase may be the rate-limiting enzyme for carbon fixation after photosynthesis.

In this study, the question of whether cytosolic FBPase can affect photosynthesis and, thus, further influence carbon allocation in the case of *N. gaditana*, an industrially used microalga, was investigated. Cytosolic FBPase was overexpressed in *N. gaditana* to explore its role in the metabolism of *N. gaditana*, further enhancing the photosynthetic carbon sequestration and biomass of the transgenic microalgal strain. The metabolites of the strain were further analyzed to explore the limiting nodes in the metabolic regulation process.

## 2. Results

### 2.1. Sequence Analysis of the cyFBPase in N. gaditana

To investigate fructose-1,6-bisphosphatase (FBPase) in different species, the sequence of the cytosolic FBPase (cyFBPase) isoform from *N. gaditana* was used as a query to search for cyFBPase homologs from diverse photosynthetic eukaryotes by utilizing BLAST (https://blast.ncbi.nlm.nih.gov/Blast.cgi; accessed on 25 May 2023) (detailed species information is provided in Appendix A). The obtained sequences were then used to construct a phylogenetic tree of cyFBPase from different organisms (Figure 1). To compare cyFBPase proteins at the amino acid level, we intentionally selected species representing significant evolutionary milestones or possessing distinctive physiological traits influencing cyFBPase activity. This selection aimed to emphasize particular amino acid structures potentially associated with variations in enzyme activity and regulation across these species, and to acquire comprehensive and high-quality cyFBPase protein sequences from public databases. We identified *Arabidopsis thaliana*, *Nicotiana tabacum*, *Chlamydomonas reinhardtii*, *Phaeodactylum tricornutum*, *Saccharomyces cerevisiae,* and others by performing a BLAST (https://blast.ncbi.nlm.nih.gov/Blast.cgi; accessed on 25 May 2023) and using *N. gaditana* as the query. The sequences were aligned and clipped using MEGA5.05 to find the best model, and, finally, the phylogenetic tree was constructed. The transmembrane domain (TMD) was predicted using InterPro (https://www.ebi.ac.uk/interpro/search/sequence/; accessed on 25 May 2023).

In this study, the genomes of 26 photosynthetic eukaryotic species representing major taxa were examined, including those with primary plastids (land plants and Chlorophyta) and secondary plastids (Bacillariophyta, Chrysophyta, Cyanobacteria, Euglenozoa, Eumycophyta, Haptophyta, Oomycota, and Pyrrophyta). We found that the *cyFBPase* of *N. gaditana* is highly homologous to other species. *cyFBPase* is prevalent in many species, not only in land plants, but also in a variety of microorganisms, including bacteria, fungi, and algae. We used EMBL-EBI (https://www.ebi.ac.uk/; accessed on 25 May 2023) for the multi-sequence alignment analysis of representative species and jalview for visualization (Appendix A). These included *A. thaliana*, *C. reinhardtii*, *N. gaditana*, *P. tricornutum*, and *S. cerevisiae*. The analysis of *cyFBPase* via InterPro (http://www.ebi.ac.uk/interpro/; accessed on 25 May 2023) revealed that the active site region (amino acids 274–286) includes a lysine residue essential for the enzyme’s catalytic mechanism. The results indicated a high similarity between the cyFBPase sequences of *N. gaditana* and *P. tricornutum*. Notably, the evolutionary tree revealed that the cyFBPase of *N. gaditana* and *C. reinhardtii* clustered into distinct evolutionary branches, suggesting distant affinities. Amino acid sequence analysis further highlighted significant differences in the cyFBPase active site of *C. reinhardtii* compared to other species, particularly *N. gaditana*. This suggests potential differential effects of cyFBPase overexpression in *N. gaditana* compared to *C. reinhardtii*.

### 2.2. Generation of a New N. gaditana Transformation Vector to Express cyFBPase from the Genome

The nucleic acid sequence of cyFBPase exhibits a high degree of conservation among various species. The *cyFBPase* gene (NCBI: Naga-100011g32) in *N. gaditana* was cloned by referencing the NCBI sequence library and subsequently searching the EST sequence library, utilizing the conserved gene sequence. The conserved fragment, spanning 1068 bp, was identified through sequence alignment. Primers were designed based on the conserved sequence to facilitate gene cloning through PCR. The PCR amplification yielded the full-length cDNA sequence of the *cyFBPase* gene, spanning 1068 bp, which was subsequently used for the construction of the overexpression vector (Appendix A). To construct transgenic strains overexpressing *cyFBPase*, target fragments were integrated into *N. gaditana* through electrotransformation. DNA from the transformed algal strains was extracted, and PCR was employed to confirm the presence of *cyFBPase* and *zeocin*-resistant fragments in the genome. Gel electrophoresis confirmed that the molecular weights of the amplified *cyFBPase* and *zeocin* fragments were approximately 1068 bp and 375 bp, respectively, which is consistent with the expected sizes (Appendix A).

The transcript levels of the algal strain overexpressing *cyFBPase* were further screened using q-PCR. We screened two overexpressing algal strains (named NgFBP1 and NgFBP2) with significantly elevated expression levels of the *cyFBP* gene. The results showed that the expression of *cyFBPase* in NgFBP1 and NgFBP2 was 35.18 and 12.40 times higher than that of the wild type (WT) (Figure 2A). The total proteins from the aforementioned algal strains were individually extracted and standardized to the same concentration for Western blot analysis. Tubulin is generally used for internal reference, as shown in Figure 2B. Notably, the overexpression of the algal strain protein exhibited a band size of 34 kDa, while no band was observed at the corresponding position in the WT strain. The above results showed that the NgFBPase overexpression plasmid was successfully inserted into the genome of *N. gaditana* and was transcribed and translated normally.

### 2.3. Analysis of cyFBPase Enzyme Activity in NgFBP Transformants

We measured the cyFBPase activity of the overexpressed strain of NgFBP using the Pi release assay and compared it with the WT strain to determine whether the transformed strain had elevated levels of cyFBPase activity, as described in the Section 4. Initially, we established the cell culture density, protein extract quantity, and FBPase reaction time needed to prepare the extracts. Subsequently, we assessed the cyFBPase enzyme activity in the algal strains overexpressing NgFBP using standardized methods. After collecting algal cells cultured until day 4 and normalizing the extracted proteins, the cyFBPase activity in the NgFBP transformants was found to be approximately 1.34 times higher than in the WT strain (Figure 3). This finding indicates that overexpressing *cyFBPase* in *N. gaditana* leads to a 1.34-fold increase in its enzymatic activity.

### 2.4. Increased Photosynthetic Characteristics of the NgFBP Overexpression Strains

The influence of *cyFBPase* gene overexpression on photosynthetic activity was further examined. To achieve this, a multi-color-pam technique was utilized to assess various chlorophyll fluorescence parameters of Photosystem II (PSII) on the third and sixth days of the light cycle, under cultivation conditions with 1.5% CO_2_ (Figure 4). The measured parameters encompassed the maximum photochemical efficiency (Fv/Fm), effective photochemical efficiency (Y(II)), and the light response curve (rETR).

The results showed an elevation in Fv/Fm values in the NgFBP overexpressing strains. At the mid-stage of the third day of cultivation, the overexpressing strains NgFBP1 and NgFBP2 displayed Fv/Fm values of 0.653 and 0.649, respectively. By the mid-stage of the sixth day of cultivation, the transformed strains exhibited Fv/Fm values of 0.657 and 0.690, respectively, whereas the WT strain showed Fv/Fm values of 0.614 and 0.647, respectively. These findings revealed a substantial increase in comparison to the WT strain (*p* < 0.05) (Figure 4A). Furthermore, the effective photosynthetic efficiency Y(II) was assessed. At the mid-stage of the third day of cultivation, the NgFBP1 and NgFBP2 exhibited Y(II) values of 0.632 and 0.631, respectively, while the WT strain had a Y(II) value of 0.612. By the mid-stage of the sixth day of cultivation, the overexpressing strains demonstrated Y(II) values of 0.569 and 0.557, respectively, whereas the WT strain had a Y(II) value of 0.525. The results demonstrated significantly higher Fv/Fm and Y(II) values in the NgFBP strains compared to the WT throughout the logarithmic growth phase (*p* < 0.05), indicating a substantially greater photosynthetic potential and improved light energy conversion efficiency in the NgFBP strains when compared to the WT (Figure 4B). The light response curve (rETR) approach is commonly employed to assess the response of *N. gaditana* to varying light intensities, examining its adaptation under different conditions. In this experiment, the relative electron transport rate (rETR(II)) of PSII was measured at the end of the logarithmic growth phase (day six), and the light response curve was generated through curve fitting. At photon intensities below 162 μmol photons m^−2^ s^−1^, the curves of the experimental and control groups exhibit close similarity. However, when the photon intensity exceeds 162 μmol photons m^−2^ s^−1^, the rETR(II) value of the overexpressing strains is significantly higher than that of the control group (*p* < 0.05).

These findings suggest that the overexpression of *cyFBP* enhances the light energy conversion efficiency of PSII and improves the responsiveness to changes in light intensity, thereby indicating a higher photosynthetic potential.

### 2.5. Effect of Overexpression of NgFBP on Growth and Biomass Accumulation during Photoautotrophic Growth

*N. gaditana* primarily accumulates organic matter through photosynthesis, and the growth and biomass accumulation of this organism directly reflect the effectiveness of photosynthesis. Therefore, this study aimed to investigate the growth and biomass production characteristics of *N. gaditana*. To assess the growth performance of the NgFBP1 and NgFBP2 lines overexpressing *cyFBP*, their growth and biomass accumulation in a seawater medium were examined.

Cell-growth-related parameters including cell number, cell size, and chlorophyll fluorescence were initially assessed using flow cytometry to compare the growth dynamics of the overexpressed strain and the WT during a 12-day culture period. Subsequently, biomass dry weight was determined on days 4 and 12 through batch cultures. Conversely, the overexpressed strains (NgFBP1, NgFBP2) exhibited higher cell numbers compared to the WT after eight days. After 12 days of culture, NgFBP1 and NgFBP2 exhibited cell numbers of 55.71 × 10^6^ cells/mL and 59.36 × 10^6^ cells/mL, respectively, which were significantly higher than the WT at 52.59 × 10^6^ cells/mL (*p* < 0.01) (Figure 5A). The average daily growth rates of the transformants NgFBP1 and NgFBP2 from day 2 to day 4 were 0.42 day^−1^ and 0.55 day^−1^, respectively, which surpassed the WT’s rate of 0.36 day^−1^. Moreover, the population overexpressing *cyFBPase* in *N. gaditana* demonstrated a significantly higher cell count than the WT (Figure 5B).

Light energy efficiency serves as an indicator of the conversion efficiency of photosynthetic organisms, transforming atmospheric CO_2_ into biomass through photosynthesis. This parameter holds significance in evaluating the effectiveness of photosynthesis in these organisms. To provide a more precise comparison of light energy efficiency under various culture conditions, the dry biomass weight of the overexpressing strains was measured on day 4 and day 12 of the batch culture (Figure 5C). The results revealed that in the batch culture until day 4, the cell dry weights of NgFBP1 and NgFBP2 were 24.24 mg/L and 21.48 mg/L, respectively, both significantly exceeding the WT at 18.95 mg/L (*p* < 0.05). At the plateau stage (day 12) of the batch culture, the cell dry weights of NgFBP1 and NgFBP2 were 250.47 mg/L and 382.47 mg/L, respectively, both significantly surpassing the WT at 214.07 mg/L (*p* < 0.05). Notably, NgFBP2 achieved nearly double the dry weight of the WT on day 12 of culture (*p* < 0.01). Hence, these findings indicate that the overexpression of *cyFBPase* in *N. gaditana* stimulates cell growth and promotes biomass accumulation.

This study revealed that NgFBP1 and NgFBP2 enhanced the total biomass per unit volume of *N. gaditana* culture and stimulated the proliferation of *N. gaditana* cells, leading to an increase in the population cell number. Biomass is determined by the mass per individual and the number of individuals, which means that besides population size, the average mass of a single *N. gaditana* cell is another factor influencing biomass. Flow cytometry was employed to assess the forward scattered light values (FSC values) of both the overexpression strains and the WT, enabling the characterization of the cell size of *N. gaditana*. The results indicated a significant (*p* < 0.05) increase in the mean single-cell FSC values on culture days 4 and 12 (Appendix A). Moreover, exoxygenic photosynthetic organisms absorb and transmit light energy through chlorophyll, which is associated with the cyst-like membrane of the chloroplast. The differences in cellular chlorophyll content were assessed in this study by measuring the chlorophyll fluorescence values of average single cells from both the overexpression strains and the WT using flow cytometry. During the logarithmic growth phase, the single-cell chlorophyll fluorescence values revealed a significant (*p* < 0.05) increase in the mean values for NgFBP1 and NgFBP2 on day 4 of the batch culture, specifically at the conclusion of the fourth photoperiod (Appendix A). In this experiment, the light energy utilization of *N. gaditana* was measured under a light intensity of 50 μmol photons m^−2^ s^−1^. The light energy utilization of NgFBP2 was 78.7% higher than that of the WT, suggesting that the overexpression of the *cyFBPase* gene enables the efficient utilization of light energy under medium light conditions (Appendix A).

The aforementioned findings demonstrate that the overexpression strains exhibited an increase in cell size, cell number, and biomass accumulation from the initial stage to day 4 of the batch culture. At the plateau stage, specifically on day 12, the overexpression strain displayed a consistent growth trend. This suggests that the overexpression of the *cyFBPase* gene exerts a substantial influence on both cell growth and biomass accumulation, accompanied by enhanced light energy utilization.

### 2.6. Lipid Content and Composition of Overexpressing Strains

For biodiesel production, it is desirable to have algal species with an appropriate lipid composition. *N. gaditana*, being a promising microalgal model for energy production, attracts considerable attention due to its cellular lipid composition and content. Consequently, we assessed the changes in the lipid composition of algal strains overexpressing *cyFBPase* to investigate its impact on lipid metabolism.

To assess the lipid yield and profile of the genetically modified strains NgFBP, we extracted oils from these algal strains overexpressing *cyFBPase* and compared their lipid yields and compositions with those of the WT. The total lipid content showed no significant variation (*p* > 0.05) between the fourth and twelfth days of incubation (Figure 6B). However, the sugar content in NgFBP exceeded that of the WT, exhibiting an average increase of 37.58% (Figure 6A). This suggests that the organic carbon assimilated through photosynthesis by the overexpressing strains of *N. gaditana* was preferentially channeled into cellular growth and division rather than lipid storage during both the logarithmic growth phase and the stationary phase.

The fatty acid composition of *N. gaditana* is primarily composed of palmitic acid (C16:0), stearic acid (C18:0), palmitoleic acid (C16:1n-7 cis), and eicosapentaenoic acid (EPA, C20:5n-3 cis). Palmitic and stearic acids are saturated, palmitoleic acid is monounsaturated, and eicosapentaenoic acid is polyunsaturated. We compared the fatty acid composition between the WT and overexpression strains. The results demonstrated a significant (*p* < 0.05) reduction in saturated (palmitic acid) and monounsaturated (palmitoleic acid) fatty acids in both overexpression strains (Figure 6C), amounting to a combined decrease of 39.64% and 43.04%, respectively, compared to the control. EPA, a polyunsaturated fatty acid, exhibited a significant increase (*p* < 0.05). In NgFBP1 and NgFBP2, the content of eicosapentaenoic acid increased by 23.08% and 20.78% of the total fatty acids (TFAs), respectively, compared to the control group. The arachidonic acid to TFA ratio in the two transformed strains of *N. gaditana* increased by 20.91% and 29.41% compared to the WT. These results indicate that the overexpression of the *cyFBPase* had no effect on the total lipid content of the cells. However, it affected the fatty acid profile by reducing the allocation of intracellular metabolic flow to carbon stores and increasing the allocation to EPA, a functional polyunsaturated fatty acid associated with vital activities.

### 2.7. Metabolomic Analysis of NgFBP Strains

To delve deeper into the post-transcriptional regulation of metabolic accumulation phenotypes resulting from NgFBP overexpression, we conducted metabolic analyses on algal strain overexpressing *cyFBPase* and the WT. Specifically, we selected NgFBP2 strains with significantly higher biomasses and EPA accumulations for metabolomic comparisons and set up six replicate samples for metabolomics assays. Principal component analysis (PCA) was applied to the samples to reflect the overall metabolic differences between them and the degree of variability within each group (Appendix A). Partial least squares (PLS) regression, a method based on covariance between predictive variables and responses, has proven effective in handling datasets with multiple collinear predictive variables (Figure 7A). In the metabolomics measurements, clear distinctions were observed between the WT (A) and NgFBP2 (B) samples, indicating significant differences in their metabolites.

The volcano plot in the study displays all substances detected in the algal strain NgFBP2 overexpressing *cyFBPase*. Each point on the plot represents a peak, with metabolites significantly increased in amount indicated in red, significantly decreased ones in blue, and metabolites without significant differences in gray. The analysis identified a total of 13,653 peaks, among which 2999 metabolites were annotated. There were 1230 metabolites with significant differences, in concentration including 663 metabolites with increased and 567 metabolites with decreased levels (Figure 7B). Correlation analysis among samples evaluates biological reproducibility within groups. The Spearman correlation coefficient, utilized as a metric for biological repeatability, indicates that R values nearing one correspond to higher reproducibility and more credible differential gene selection. The results demonstrate robust biological reproducibility in identical samples, whereas the correlations are notably lower in different treatments, affirming strong experimental repeatability (Appendix A).

The metabolic profiling and KEGG enrichment analysis of differentially abundant metabolites (DAMs) the top-20 enriched pathways (Appendix A). The enriched metabolic pathways mainly include metabolic pathways related to glycine, serine, and threonine metabolism, glycerophospholipid metabolism, pantothenate and CoA biosynthesis, porphyrin metabolism, and carotenoid biosynthesis (Figure 7C). Further analysis of DAMs in potential important metabolic pathways revealed that the upregulation of these DAMs mainly included protoporphyrin, (3Z)-phycocyanobilin, beta-zeacarotene, astaxanthin, and alpha-carotene (Figure 7D). The results indicated that heightened cyFBPase activity might boost photosynthesis and gap junction communication by modulating the levels of porphyrin signaling molecules or through the biosynthesis of carotenoid-related compounds, thus fostering algal cell proliferation.

## 3. Discussion

### 3.1. The Activity of cyFBPase in Photosynthesis and Its Effects

In this research, we isolated the complete cDNA sequence from *N. gaditana*, which encodes a protein of 356 amino acids. Phylogenetic analyses show that the cyFBPase from *N. gaditana* fits well within the algal FBPase family, sharing closer structural characteristics with *P. tricornutum* than with *C. reinhardtii*, suggesting potential functional variations. Typically, the algal cytoplasmic cyFBPase activities are substantially lower than those of other enzymes in the same cycle [32]. For instance, in *E. gracilis* cells, the activity of FBPase is lower than that of NADP+-dependent glyceraldehyde-3-phosphate dehydrogenase and phosphoribulokinase [33]. Under conditions of low light, the rate-limiting step in photosynthetic CO_2_ assimilation is electron transport rather than the capacity of the Calvin cycle. At light intensities below 100 μmol·m^−2^·s^−1^, transformed strains show improved relative electron transfer rates [34]. However, the specific substances that are crucial for enhancing photosynthesis and growth under photoautotrophic conditions in these genetically modified microalgae are not well defined [2,35].

Through genetic engineering, cyFBPase activity was increased in *N. gaditana*, which led to enhanced photosynthesis and growth. The metabolomic results linked this to increased porphyrin metabolism and carotenoid-biosynthesis-related metabolite accumulation, affecting carbon allocation. In many plant studies, porphyrin metabolism and carotenoid biosynthesis directly affect photosynthetic efficiency [36,37]. Consequently, significant changes in protoporphyrin concentrations were observed in our transgenic microalgae. These changes suggest that protoporphyrins, as key players in photosynthesis, significantly influence their own levels [25,38]. Additionally, protoporphyrins may act as intermediates, converting into other substances like sugars. The intertwined pathways of porphyrin biosynthesis and sucrose metabolism in plant cells indicate that sucrose may act as a regulatory molecule that controls the synthesis of chlorophyll and porphyrins essential for photosynthesis [25,38,39,40]. The degradation of sucrose also supplies the necessary precursors and energy for porphyrin production [41]. Overall, *cyFBPase* in *N. gaditana* is a key regulatory point in algal cell growth. The functional analysis of *cyFBPase* provides insights into the evolution of metabolic pathways and lays the groundwork for potential industrial uses for microalgae.

### 3.2. Transforming Microalgae through Engineering to Surpass the Constraints of Photoautotrophic Efficiency

In higher plants, multiple factors can diminish photosynthetic efficiency. These factors include the decreased efficiency of electron transfer due to structural flaws, dissipation of the majority of absorbed light energy as heat because of overly long light-absorbing antennae, a narrow effective spectral range for utilizing light, and reduced rates of carbon assimilation during the dark reaction phase. As a result, only a small portion of sunlight is transformed into solar energy, leading to photosynthetic efficiencies that are much lower than theoretical expectations [42]. Enhancements to the photosynthetic system have significantly improved photosynthetic performance and growth in plants. For example, after overexpressing the *FBPase* gene, wheat exhibited marked increases in both total biomass and dry seed yields [43]. Although the photosynthetic mechanism in *N. gaditana* remains partially undefined, with potential flaws needing more exploration, targeted enhancements of certain aspects of its photosynthetic system have effectively improved both its photosynthetic efficiency and biomass production. The upregulation of photosynthesis-related genes, such as *psbO*, *petC*, *petF*, and *petH*, in a modified strain of *N. gaditana* resulted in a 26.7% increase in biomass productivity relative to the WT [44]. Genetic engineering, an effective method for adaptive breeding, is extensively used, and *N. gaditana*, with its straightforward genome and the public availability of its genomic information online, has become a prime candidate [45]. Therefore, the precise genetic alteration of essential transcription factors, enzymes, and protein subunits that play roles in photorespiration and the Calvin cycle in *N. gaditana* offers a hopeful strategy for surpassing the photosynthetic constraints of this species [46].

## 4. Materials and Methods

### 4.1. Strains and Culture Conditions

*Nannochloropsis gaditana* CCMP 526 was a kind gift from Danxiang Han (Institute of Hydrobiology, Chinese Academy of Sciences). NgFBP, a strain that overexpresses the cyFBPase, was obtained through electroporation. In addition, NgFBP is resistant to zeocin, and the screening concentration was 2 µg·mL^−1^.

*N. gaditana* was inoculated into artificial seawater (Fauna Marin, Holzgerlingen, Germany) supplemented with f/2 medium (catalog number G0154, Sigma-Aldrich, St. Louis, MO, USA) and cultured in 175 cm^2^ breathable sterile square-cell culture flasks (Corning^®^, Corning, NY, USA) containing 100 mL of the culture medium. The initial concentration of *N. gaditana* was 1 × 10^6^ cells/mL. Each treatment was set up in triplicates and incubated in a CO_2_ light incubator with a CO_2_ concentration of 1.5% of air volume, a light intensity of 50 μmol m^−2^ s^−1^, a 12 h light/dark cycle, and a temperature of 22 °C.

### 4.2. Cloning Full-Length Genes and Construction of Overexpression Vectors

Specific primers were designed based on the *cyFBPase* sequence, and the resulting PCR amplification using the following primer (forward primer 5′-cacactctaaaccccaataaaatggcgactaaagacatcg-3′ and reverse primer 5′-gtcgtcatccttgtaatccgccaaggccaactctttc-3′). The PCR amplification yielded the full-length cDNA sequence of the *cyFBPase* gene, spanning 1068 bp, which was subsequently used for the construction of the overexpression vector. The amplified fragment mentioned above was Gibson-ligated and transformed into *E. coli* using the pHSP plasmid as the backbone, which carries the pHsp20 promoter and zeocin resistance [47].

### 4.3. Electrotransformation, DNA Extraction, and PCR Detection

To generate a transgenic strain with overexpression of the *cyFBPase*, the target fragment was amplified from the plasmid and then integrated into *N. gaditana* through electrotransformation. The electroshock parameters were 2200 V, 50 μF and 600 Ω [13]. After electroshock, the algal cells were transferred to centrifuge tubes containing 10 mL of f/2 medium (catalog number G0154, Sigma-Aldrich, USA) and incubated under low light at 22 °C for 24–48 h. After centrifugation at 4000× *g* for 10 min, the supernatant was removed and spread evenly on resistant plates supplemented with 2 μg/mL zeocin. The plates were incubated at 22 °C, a light intensity of 50 μmol m^−2^ s^−1^, and a 12 h light/dark cycle.

Total DNA was extracted from the samples using a high-efficiency plant genomic DNA extraction kit (GeneBetter^®^, Beijing, China). Gene-specific primers (Table 1) were used to verify the PCR amplification of the *cyFBPase* and resistant zeocin fragments derived from the transformed *N. gaditana* genome. The PCR program was initiated at 94 °C for 30 s, followed by 35 cycles of 98 °C for 10 s, 55 °C for 30 s, and 72 °C for 1 min. The PCR products were visualized using electrophoresis on 1% (*w*/*v*) agarose gels.

### 4.4. Cell Growth and Biomass Measurements

The cell growth was analyzed by measuring the cell density (in cells/mL) and dry cell weight (DCW). Cells were inoculated at 106 cells/mL, determined using CytoFLEX S flow cytometer(Beckman, Brea, CA, USA). The sample was quantified via CytoFLEX S flow cytometer(Beckman, Brea, CA, USA) using a fluorescence channel with excitation light at 488 nm and emission light at 685 nm to quantify the concentration of *N. gaditana* CCMP526 cells. For assessing the particle size and chlorophyll fluorescence of algal cells, sample data collection was conducted with a CytoFLEX S flow cytometer (Beckman, Brea, CA, USA), utilizing CytExpert 2.5 software. A 488 nm laser enabled cell particle size measurements via the FSC-A and SSC-A channels, while the 690 nm/50 BP channel facilitated the analysis of chlorophyll fluorescence to determine the chlorophyll content at the single-cell level.

At the end of the photoperiod on day 12 of the culture, the same volume of the transformed algal strain as the control *N. gaditana* culture was collected by centrifugation at 5000× *g* for 10 min. The algae were washed three times through centrifugation in deionized water to remove the salt, collected by centrifugation in clean, dry centrifuge tubes that had been weighed, and placed in a vacuum drying oven set at 105 °C for 24 h. After drying, the dry weights (accurate to 0.1 mg) were weighed on an electronic balance (Sartorius, Quintix^®^, Gottingen, Germany).

### 4.5. Calculations of Light Energy Utilization

The light energy utilization of *N. gaditana* was calculated using an equation (PCE = DW × Q/(PAR × T × S × E)) [48]. The formula for light energy utilization was derived from sources in the literature, and the calorific value of microalgae biomass was estimated at 29 KJ/g based on the proportions of the three major nutrients. In this formula, DW represents the dry weight growth of photosynthetic organisms over a given period (g), Q denotes the heat of combustion of biomass (KJ/g), PAR indicates the photosynthetically active radiation (mol m^−2^ s^−1^), S represents the light-exposed area (m^2^), T stands for the duration of light exposure (s), and E signifies the average energy of light within the 400–700 nm wavelength range (KJ/mol).

### 4.6. Chlorophyll Fluorescence Analysis

A multi-excitation wavelength-modulated chlorophyll fluorometer (MULTI-COLOR-PAM, Walz, Germany), alongside the PAMWin3 software, facilitated the measurement of chlorophyll fluorescence parameters in *N. gaditana*. We primarily assessed the following PSII chlorophyll fluorescence parameters: maximum photosynthetic efficiency (Fv/Fm), actual photosynthetic efficiency (Y(II)), and the relative electron transfer rate (rETR(II)). Selected algal samples from the batch culture were moderately diluted to achieve a chlorophyll content of approximately 200 μg/L and exposed to light for 20 min. Prior to measurement, the instrument was calibrated to zero. The samples were then placed in a cuvette, with the Ft value adjusted to 1.5 before initiating the measurement process. The measuring light (ML) was set at 440 nm, and white light was chosen as the photochemical light (AL), matching the intensity used during incubation. The operational procedure was as follows: a total of 1.5 mL of the sample was introduced into a cuvette with a 10 mm diameter to measure the chlorophyll fluorescence of *N. gaditana*. Initially, the minimum fluorescence (Fo) under dark conditions was recorded using the measuring light. Subsequently, the maximum fluorescence (Fm) was determined by activating the saturating flash at an intensity of 3000 μmol photons m^−2^ s^−1^. For light acclimation, the photochemical light was turned on, with saturating pulses spaced by 20 s intervals. During this process, the steady-state fluorescence (F) and the maximum light-adapted fluorescence (Fm′) were recorded. The calculations for maximum photosynthetic efficiency (Fv/Fm) and actual photosynthetic efficiency (Y(II)) employed the formulas Fv/Fm = (Fm − Fo)/Fm and Y(II) = (Fm′ − F)/Fm′.

The relative electron transfer rate (rETR) and the fast photoresponse curve of Photosystem II were determined by setting the measurement light wavelength to 440 nm, setting 20 steps (0, 2, 3, 14, 26, 38, 57, 82, 113, 162, 217, 276, 348, 418, 505, 607, 713, 833, 999, and 1172 μmol m^−2^ s^−1^). White light at the wavelength of 420 ~ 640 nm was used as the photochemical light, with a duration of 20 s per shift and a saturation pulse (duration of 800 ms) turned on at the end of each shift. The relative electron transfer rate (rETR) for each light intensity was determined using the equation rETR(II) = Y(II) × PAR × 0.5. Here, Y(II) represents the effective photon yield from the equation, PAR denotes the photochemical light intensity (in μmol photons m^−2^ s^−1^), and the factor 0.5 assumes an equal distribution of absorbed photons between Photosystem I and II. The rETR values were modeled based on the equation rETR = PAR/(a × PAR^2^ + b × PAR + c), where PAR is the specified photochemical light intensity, and a, b, and c are fitting parameters. Subsequently, from the light response curves, both the maximum relative electron transfer rate (rETRmax) and the light saturation point (Ik) were derived using Eilers and Peeters (1998) equations: rETRmax = 1/(b + 2 × (a × c)^1/2^) and Ik = c/(b + 2 × (a × c)^1/2^) [49,50].

### 4.7. RNA Extraction and Quantitative Real-Time PCR

The microalgal samples were collected using 0.4 μM PCM (Millipore, IsoporeTM, Billerica, MA, USA) filtered through an extraction flask (Millipore, Strifil^®^, Billerica, MA, USA), eluted with 2 mL PBS buffer into a 2 mL grinding tube, and centrifuged at 5000× *g* for 5 min at 4 °C. The supernatant was discarded and then immediately frozen in liquid nitrogen. The frozen algae and pre-cooled sterile steel beads were placed in a freezer grinder for 30 s at 55 Hz. After grinding was complete, the total RNA was extracted from the samples using the polysaccharide/polyphenol plant Total RNA Mini Kit (GeneBetter^®^, Beijing, China). The expression of *NgFBP* and the target genes was quantitated with qRT-PCR. We analyzed the expression of *NgFBP* using primers qFBP-F and qFBP-R (Table 2). The PCR program was initiated at 95 °C for 30 s, followed by 40 cycles of 95 °C for 5 s and 60 °C for 30 s. mRNA levels were quantified using the 2^−ΔΔCT^ method [51]. The housekeeping *actin* gene was used as a loading control.

### 4.8. Western Blotting

Overexpressing and WT algal cells in the mid-exponential phase were used and lysed by adding 200 μL of cell lysis solution (P0013, Beyotime, Shanghai, China) and 1 mM of the protease inhibitor PMSF. Total protein was obtained via centrifugation (4 °C, 16,000 rpm, 20 min) after lysis. Protein was quantified using the BCA method (P0010SN, Beyotime, Shanghai, China) and analyzed using SDS-PAGE (12%TGX Fase Case Kit, Bio-Rad, Hercules, CA, USA). After electrophoresis, the membrane was transferred (PVDF) at 100 V for 1 h. The membrane was washed 3 times in 1 × TBST (100 mM Tris-HCl, 150 mM NaCl, 0.1% Tween-20), and then incubated for 1 h at 37 °C in a blocking solution (5% skimmed milk, diluted with 1 × TBST). After blocking, the samples were each washed 3 times for 10 min in 1 × TBST. The expression of FLAG-tagged NgFBP was confirmed using the rabbit anti-FLAG-tag antibody (LABLEAD, Beijing, China) at a dilution of 1:1000, followed actin anti-rabbit secondary antibody (LABLEAD, Beijing, China) at a dilution of 1:2000. Signals were detected using enhanced chemiluminescence (ECL) and the ChemiDoc system (Bio-Rad, Hercules, CA, USA).

### 4.9. Lipid Quantification

Total lipids were extracted starting from lyophilized samples. The material was shredded as much as possible to create a powder that was easy to weigh and to facilitate lipid extraction. After grinding, 6 mL of a mixture of methanol, chloroform, and formic acid (20:10:1 by volume) was added to the extract, which was then vortexed and shaken for 10 min. Then, 3 mL of a mixture of phosphoric acid and potassium chloride (0.2 M H_3_PO_4_, 1 M KCl) was added, and the sample was vortexed and shaken for 2 min before being centrifuged at 5000× *g* for 10 min. The chloroform layer was pipetted into a 10 mL tube and blow-dried with nitrogen. The total lipid content of the sample was determined according to the following formula: total lipid content (% DW) = A/DW (A is the total lipid mass of the sample; DW indicates the dry weight of the sample).

### 4.10. cyFBPase Enzyme Assay

The cyFBPase activity in the WT and transformed strains was quantified using a colorimetric assay. This involved measuring the increase in the absorbance of malachite green at 620 nm, indicative of phosphate release by cyFBPase from fructose-1,6-bisphosphate, as detailed in the Bioassay Systems© 2009 manual [22]. Briefly, cells from 20 mL cultures of WT and various NgFBP transgenic lines, grown to a density of 3 × 10^7^ cells/mL, were collected via centrifugation at 3000× *g*. The cells were then pulverized in liquid nitrogen, lysed with 200 μL of cell lysis solution (P0013, Beyotime, Shanghai, China) and 1 mM PMSF protease inhibitor, and centrifuged at 4 °C, 16,000 rpm for 20 min to extract the total protein.

Enzymatic reactions were induced by adding 15 μL of extract (optimized from initial tests ranging 5–20 μL) to 80 μL of assay buffer (50 mM Tris, pH 8.2, 15 mM MgCl_2_, 1.5 mM EDTA, 10 mM DTT, 2 mM FBP) and incubating at 25 °C for 5 min (determined as the optimal time). The reactions were halted by adding 50 μL of 1 M perchloric acid, followed by centrifugation at 11,000× *g* for 10 min. An 80 μL aliquot from each reaction was transferred to a 96-well plate, to which 20 μL of malachite green reagent (reagent A: 0.20% malachite green, 4% ammonium molybdate, 12% sulfuric acid; reagent B: 15% polyoxyethylenesorbitan monolaurate, mixed at a 100:1 ratio) was added. Following a 30 min room-temperature incubation, absorbance at 620 nm was recorded using a plate reader. Activity was then calculated based on the absorbance readings and a standard [Pi] curve, as per the BioAssay Systems© protocol (2009).

### 4.11. Fatty Acid Composition Analysis (GC-MS)

Samples were placed into 2 mL EP tubes, extracted with 500 μL extracting solution (VIsopropanol/Vn-Hexane = 2:3, containing 0.2 mg/L internal standard), and vortexed for 30 s. They were then homogenized in a ball mill for 4 min at 40 Hz, and then ultrasound-treated for 5 min (incubated in ice water) before being centrifuged for 15 min at 12,000 rpm at 4 °C. The supernatant was then transferred into fresh 2 mL EP tubes. Then, 500 μL of extract solution (VIsopropanol/Vn-Hexane = 2:3, containing 0.2 mg/L internal standard) was added to the remaining samples, which were vortexed for 30 s. A total of 800 μL of the combined supernatant was taken and placed into a freeze centrifuge concentrator for drying. Following this, 500 μL of methanol/trimethylsilyl diazomethane solution (1:2) was kept at room temperature for 30 min and then blow-dried using nitrogen. Then, 160 μL of n-Hexane was added and redissolved before being centrifuged for 1 min at 12,000 rpm [52]. The supernatant was transferred into a fresh vial for GC-MS analysis.

GC-MS analysis was performed using an Agilent 7890B gas chromatograph system coupled with an Agilent 5977B mass spectrometer. The system utilized a DB-Fast FAME capillary column. A 1 μL aliquot of the analyte was injected in split mode (5:1). Helium was used as the carrier gas; the front inlet purge flow was 3 mL min^−1^, and the gas flow rate through the column was 46 psi with constant pressure. The initial temperature was held at 50 °C for 1 min; increased to 200 °C at a rate of 50 °C min^−1^ and held for 15 min; increased to 210 °C at a rate of 2 °C min^−1^ and held for 1 min; and increased to 230 °C at a rate of 10 °C min^−1^ and held for 15 min. The injection, transfer line, quad, and ion source temperatures were 240 °C, 240 °C, 230 °C, and 150 °C, respectively. The energy was −70 eV in electron impact mode. The mass spectrometry data were acquired in Scan/SIM mode with the *m*/*z* range of 33–400 after a solvent delay of 7 min [53]. The fatty acid composition was calculated as a percentage of the total fatty acids present in the sample.

### 4.12. Metabolomic Analysis

The sample was transferred to an EP tube with 1000 μL of the extract solution (methanol acetonitrile *v*/*v* = 1:1, internal standard concentration 20 mg/L) containing the internal standard in three batches (300 μL, 300 μL, 400 μL), vortexed, and mixed for 30 s. After grinding and sonication, the extract was shaken for 10 min at 45 Hz, and centrifuged for 15 min at 12,000 rpm at 4 °C. Then, 500 μL of the supernatant was carefully removed from the EP tube and the extract dried in a vacuum concentrator; 160 μL of extract solution (acetonitrile/water *v*/*v*) was then added to the dried metabolite, vortexed for 30 s, and sonicated for 10 min in an ice water bath. The sample was centrifuged at 4 °C at 12,000 rpm for 15 min, then 120 μL of the supernatant was carefully removed and placed in a 2 mL injection bottle, and, finally, 10 μL of each sample was mixed to form a QC sample for testing. The LC/MS system for metabolomics analysis is composed of a Waters Acquity I-Class PLUS ultra-high-performance liquid tandem and a Waters Xevo G2-XS QTof high-resolution mass spectrometer. The column used was a Waters Acquity UPLC HSS T3 column (1.8 μm 2.1 × 100 mm). The mobile phase consisted of 0.1% formic acid aqueous solution and 0.1% formic acid acetonitrile. The injection volume was 1 μL. The Waters Xevo G2-XS QTof high-resolution mass spectrometer can collect primary and secondary mass spectrometry data in MSe mode under the control of the acquisition software (MassLynx V4.2, Waters, Milford, CT, USA). In each data acquisition cycle, a dual-channel data acquisition can be performed on both low-collision energy and high-collision energy at the same time. The low-collision energy is 2 V, the high-collision energy range is 10 ~ 40 V, and the scanning frequency is 0.2 s for a mass spectrum. The parameters of the ESI ion source are as follows: capillary voltage, 2000 V (positive ion mode) or −1500 V (negative ion mode); cone voltage, 30 V; ion source temperature, 150 °C; desolvent gas temperature, 500 °C; backflush gas flow rate, 50 L/h; and desolventizing gas flow rate, 800 L/h [54]. The raw data collected using MassLynx V4.2 were processed by Progenesis QI software 2.0 for peak extraction, peak alignment, and other data processing operations, based on Progenesis QI software 2.0’s online METLIN database and Biomark’s self-built library for identification. The theoretical fragment identification and mass deviation were all within 100 ppm.

### 4.13. Statistical Analysis

Biological and technical replicates, along with the statistical analyses of the data generated in this study, followed the following methodology: Each growth condition was analyzed using two biological replicates, supported by three technical replicates for each, to validate the reproducibility of the observed trends and relationships. Averages from these technical replicates were determined for every data point. This approach was consistently applied in the enzyme activity assays and related experiments. Data from all technical replicates for each biological replicate were consolidated for ANOVA analysis. Error bars represent the standard deviation from the mean of triplicate samples. Treatment effects were evaluated using Tukey’s test (*p* < 0.05). All statistical analyses utilized GraphPad Prism 8.4 (GraphPad Software, 2020, Boston, MA, USA), employing two-way ANOVA.

## 5. Conclusions

This study shows that overexpressing cyFBPase in *N. gaditana* significantly boosts photosynthesis and biomass in transgenic lines. It increases the proportion of unsaturated fatty acids and activates metabolic pathways linked to glycine, protoporphyrin, and auxin energy production. As a result, there is a shift in the carbon skeleton from glycine to protoporphyrins and carotenoids, fostering metabolic processes associated with chloroplast development. Metabolomic analyses provide initial insights into the mechanisms driving accelerated photosynthetic energy conversion and enhanced growth. Additionally, the proportion of EPA, a valuable product in *N. gaditana*, significantly rises in medium fatty acids, setting the stage for the further enhancement of EPA production through genetic engineering. Our future research will focus on optimizing the growth conditions and metabolic regulation pathways of NgFBP2 to increase biomass and lipid productivity, facilitating the development of products such as biodiesel. Advances in molecular biology technology, along with improved genomic data and the genetic manipulation tools for *N. gaditana*, offer technical support for its use as a cell factory for producing energy, high-value active products, and proteins through genetic modification. We will incorporate other strategies, such as improving the efficiency of light energy conversion, to develop algal strains with superior traits in terms of the biomass and yields of target products.

## Figures and Tables

**Figure 1 ijms-25-06800-f001:**
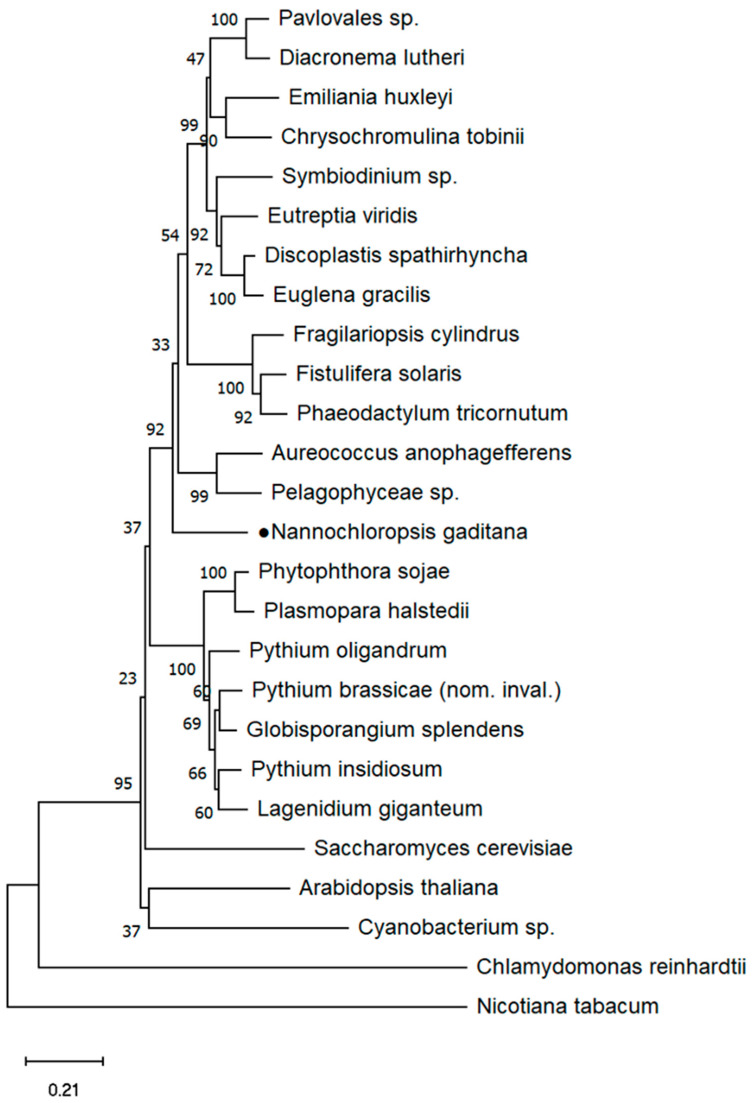
Phylogenetic tree of gene *cyFBPase* from different photosynthetic eukaryotic species. Black dot represent highlighting.

**Figure 2 ijms-25-06800-f002:**
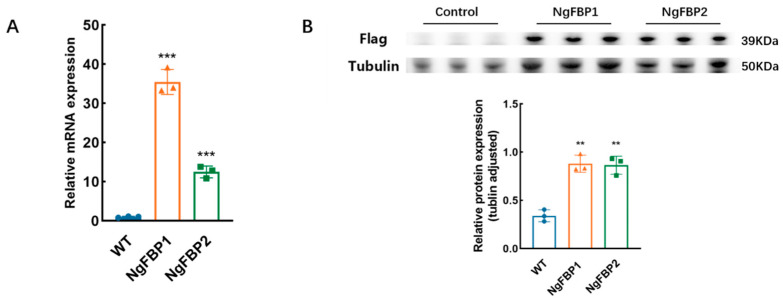
Validation of transcriptional and protein expression levels in NgFBP transformants. (**A**) Gene expression levels of *cyFBPase* between WT and NgFBP transformants determined through RT-qPCR. (**B**) Expression levels of cyFBPase between WT and NgFBP transformants determined through Western blotting. ** *p* < 0.01, *** *p* < 0.001. The error bars indicate the standard deviation from three replicate calculations. All data were analyzed through two-way ANOVA (GraphPad Prism).

**Figure 3 ijms-25-06800-f003:**
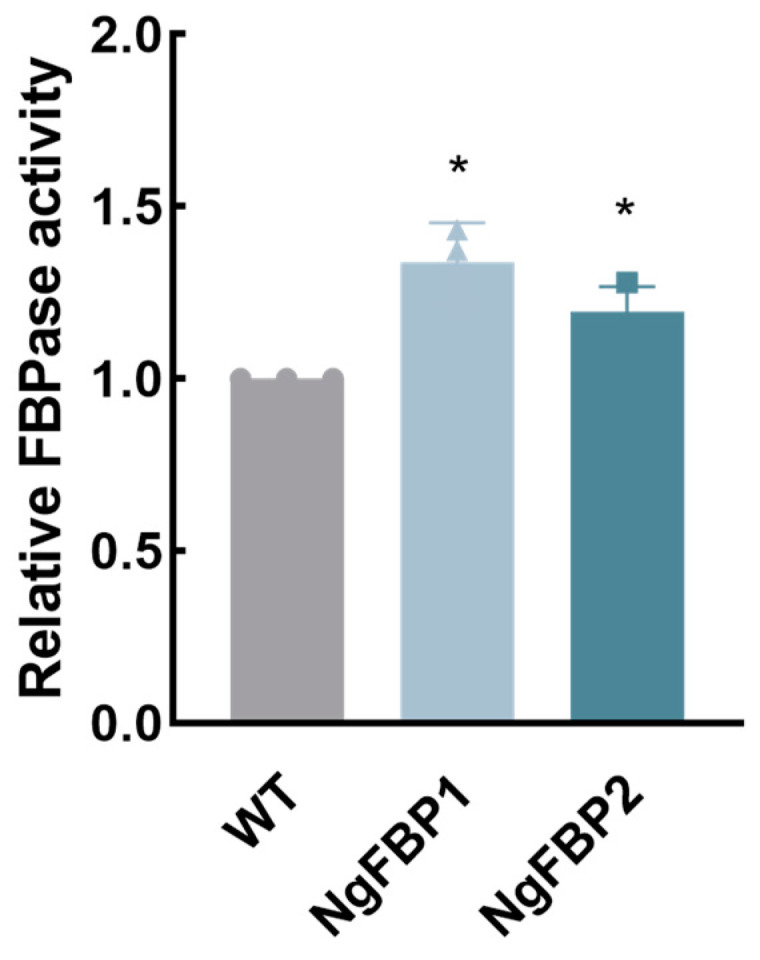
Relative cyFBPase enzyme activity in NgFBP. * *p* < 0.05. The error bars indicate the standard deviation from three replicate calculations. All data were analyzed through two-way ANOVA (GraphPad Prism).

**Figure 4 ijms-25-06800-f004:**
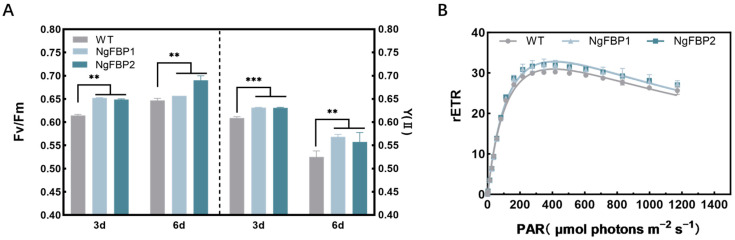
Chlorophyll fluorescence parameters of Photosystem II (PSII) of WT and NgFBP strains. (**A**) Maximum photochemical efficiency (Fv/Fm) and effective photochemical efficiency (Y(II)). (**B**) Light response curve (rETR). ** *p* < 0.01, *** *p* < 0.001. Error bars represent standard deviations calculated from three independent biological replicates. All data were analyzed through two-way ANOVA (GraphPad Prism).

**Figure 5 ijms-25-06800-f005:**
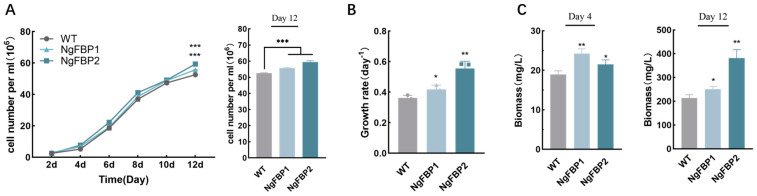
Growth curve, growth rate, and biomass accumulation of NgFBP strains. (**A**) Growth curve of WT and NgFBP after inoculation and number of cells cultured on day 12. (**B**) Growth rate from day 4 to day 6. (**C**) Biomass of WT and NgFBP on day 4 and day 12. * *p* < 0.05, ** *p* < 0.01, *** *p* < 0.001. Error bars indicate standard deviations. Cell density, growth rate, and biomass were analyzed using two-way ANOVA (GraphPad Prism).

**Figure 6 ijms-25-06800-f006:**
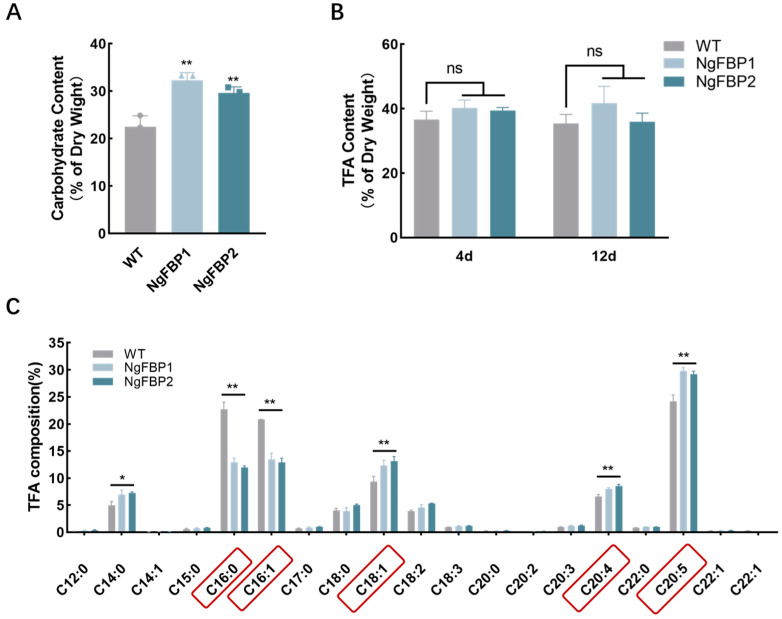
Biochemical composition of WT and NgFBP strains. (**A**) Carbohydrate contents on day 12. (**B**) TFA contents on day 4 and day 12. (**C**) Fatty acid composition of WT and NgFBP on day 12. ns indicates *p* > 0.05, * *p* < 0.05, ** *p* < 0.01. Error bars represent standard deviations calculated from three independent biological replicates. Red frames indicate significantly changed components. All data were analyzed through two-way ANOVA (GraphPad Prism).

**Figure 7 ijms-25-06800-f007:**
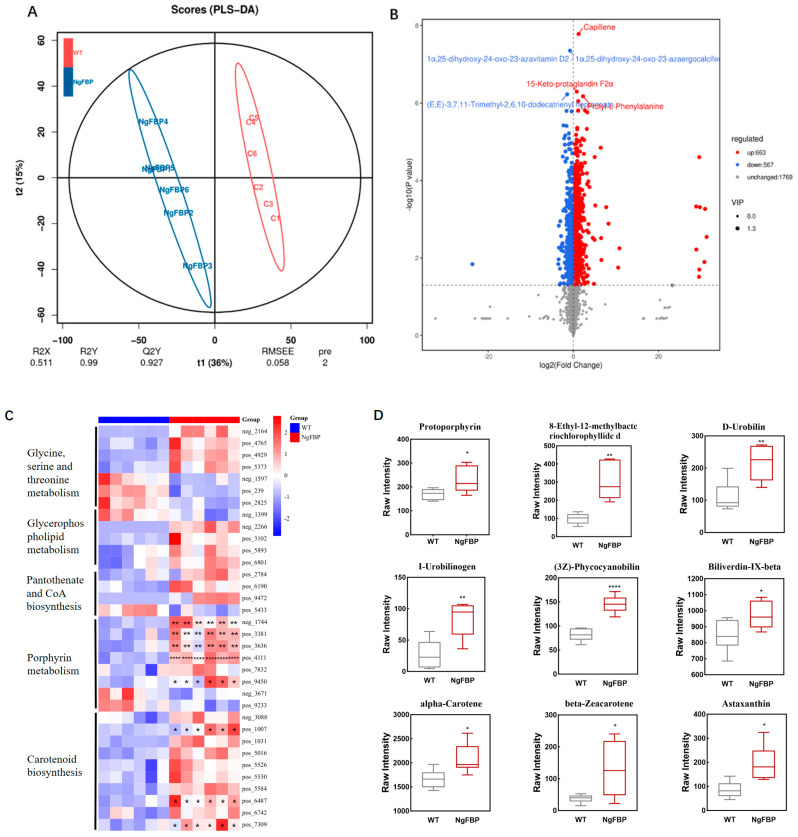
Metabolomics of the NgFBP2 strain. (**A**) PLS-DA analysis of WT and NgFBP2 strains. (**B**) Volcano plot of WT and NgFBP2 strains. (**C**) Metabolite heatmaps of differentially abundant metabolic pathways. (**D**) Metabolites with major increase in porphyrin and carotenoid metabolism. * *p* < 0.05, ** *p* < 0.01, **** *p* < 0.0001. Error bars represent standard deviations calculated from six biological replicates. All data were analyzed through two-way ANOVA (GraphPad Prism). The percentages listed in the coordinate axis labels describe the proportions of variance explained by the first (R_2_X) and second (R_2_Y) principal components, respectively. Colors denote the metabolite’s abundance level. Red signifies an increase, while blue indicates a decrease.

**Table 1 ijms-25-06800-t001:** Primer sequences used in PCR.

Primer Name	Primer Sequences
PcyFBP-1F	GGCTGGCAATGTTCTGTTTG
PcyFBP-1R zeo-1F zeo-1R	CTGGTTGAGTTCGATAGCAC ATGGCCAAGTTGACCAGTGC GGTTCAGTCCTGCTCCTCGG

**Table 2 ijms-25-06800-t002:** Primer sequences used in qRT-PCR.

Primer Name	Primer Sequences
qFBP-F	GCGGTGCTTGTGTCTGAGGAA
qFBP-R Actin-F Actin-R	GCTCGTAGATGGCGAAGATGGT AGCTGCCGGATGGTAACGTG GCTCGCCTCCTTGCCGATAA

## Data Availability

The raw data supporting the conclusions of this article will be made available by the authors on request.

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
