# Peer review of "Metabolomics Reveals the Impact of Overexpression of Cytosolic Fructose-1,6-Bisphosphatase on Photosynthesis and Growth in Nannochloropsis gaditana"

_ijms, 2024, doi:10.3390/ijms25126800_

Round 1
Reviewer 1 Report (New Reviewer)
Comments and Suggestions for Authors
Review Report
Reviewing report regarding the manuscript entitled ‘Metabolomics reveals the impact of overexpression of cytosolic fructose-1,6-bisphosphatase on Photosynthesis and Growth in Nannochloropsis gaditana’
- Please change the title of your manuscript to match the great work you did in your research.
- The manuscript did not include recent references for 2023 and 2024.
- Please rewrite the abstract and make sure that the sequence of your abstract fits with your manuscript.
- The authors are using the word ‘recent’ many times even if the citation of this recent work goes back to 2014 and 2018. So, please make sure that you use the right words that match your citations.
- Introduction too long
- Please rewrite your introduction and also your aim of research.
Results
- Please describe your result only and avoid describing your methodology in the results section.
- The title of your Figure (1) does not match the figure itself, so please change it accordingly.
- You can move Figure 1 B in the supplementary file.
- Please move Figures 2 A and B to the supplementary file.
- Please rewrite the title of Figure 3 and name the Y axes.
- Please rewrite the title of Figure 5 and change the explanation for each figure in Figure 4.
- Please add a title for the Y axes in Figure 6.
Discussion
- Please rewrite the discussion.
Material and methods
- Please make sure that you briefly describe your methodology and cite it in the end.
Conclusion
- Make sure that your conclusion focuses on the importance of your work and your research message plus prospects.
Comments on the Quality of English LanguageAuthor Response
Dear reviewer,
Thank you for reviewing our manuscript ID: ijms-3021227, which entitled "Utilizing metabolomics to elucidate the effects of cytosolic fructose-1,6-bisphosphatase overexpression on photosynthesis and growth in Nannochloropsis gaditana.", and providing additional comments and recommendations, which have not only made this work scientifically more accurate, but also helped to improve its readability to a broad audience. Our revised manuscript has been greatly improved according to the comments. All questions and suggestions were addressed and corrected accordingly. Please see the attachment.
Thank you and best regards.
Sincerely,
Hantao Zhou
State Key Laboratory of Marine Environmental Science, Xiamen University
College of Ocean and Earth Sciences, Xiamen University
Xiamen, Fujian 361000, China
E-mail: htzhou@xmu.edu.cn

Reviewer 2 Report (New Reviewer)
Comments and Suggestions for Authors
The manuscript submitted by Zhang et al. investigated the overexpression of the cy FBPase enzyme in microalgae Nannochloropsis gatidana, with a particular focus on the effect on biomass production and photosynthetic activity. The work is correctly structured, and the principal effects were subjected to thorough investigation and detailed description.
Main considerations:
Line 107: Please correct the expression: Overexpression cytosolic FBPase OF Brassica napus instead of overespression cytosolic FBPase in Brassica napus
Line 160: Naga-100011g32? What is? Please specify better.
Line 172: please insert the reference for the pHSP plasmid
Figure 2B: The size of the cyFBPase gene is 1068 bp, as specified in the text. Upon examination of figure 2b, however, the size appears to be larger in comparison to the molecular marker utilized.
In the same figure (2B), it is unclear what the NC designation signifies. Is it a negative control? If so, why does it amplify for zeacin?
Line 187: Please rephrase the following sentence: The results (Figure 2-D) indicated that tubulin antibody served as the internal reference. Tubulin is generally used for internal reference!
Line 563 4.4 RNA extraction and Quantitative real-time PCR
Please confirm whether the primers used for quantitative PCR were sourced from the literature. If this is the case, please provide the relevant reference. Alternatively, if they were designed in-house, please confirm whether they have been validated and their efficiency checked. Please insert the qPCR amplification conditions in the text.
RNA extraction kit (Genebetter ?)
Line 581: It is important to note that the concentration of Tween should be indicated, rather than the quantity.
Line 583:
Please provide details of the concentration and type of secondary antibody employed.
Line 681: indicate version and year of GraphPad Prism software.
Author Response
Dear reviewer,
Thank you for reviewing our manuscript ID: ijms-3021227, which entitled "Utilizing metabolomics to elucidate the effects of cytosolic fructose-1,6-bisphosphatase overexpression on photosynthesis and growth in Nannochloropsis gaditana.", and providing additional comments and recommendations, which have not only made this work scientifically more accurate, but also helped to improve its readability to a broad audience. Our revised manuscript has been greatly improved according to the comments. All questions and suggestions were addressed and corrected accordingly. Please see the attachment.
Thank you and best regards.
Sincerely,
Hantao Zhou
State Key Laboratory of Marine Environmental Science, Xiamen University
College of Ocean and Earth Sciences, Xiamen University
Xiamen, Fujian 361000, China
E-mail: htzhou@xmu.edu.cn

Round 2
Reviewer 1 Report (New Reviewer)
Comments and Suggestions for Authors
The Authors changed the MS according to the comments.
Author Response
Thank you for your comments, we have incorporated them into our manuscript
Reviewer 2 Report (New Reviewer)
Comments and Suggestions for Authors
Dear authors,
Thanks for your kindly reply.
Author Response
Thank you for your comments, we have incorporated them into our manuscript
This manuscript is a resubmission of an earlier submission. The following is a list of the peer review reports and author responses from that submission.
Round 1
Reviewer 1 Report
Comments and Suggestions for Authors
1. According to my understanding, the native gene is being overexpressed so can the authors explain why there is not FBpase band in WT? Additionally, if there is no band representing this enzyme then even the protein level should be negligible right? is this protein known to be very stable and low turnover?
2. Figure 3: How significantly different is the enzyme activity in FBP1 and 2? I am guessing its non-significant. If thats the case then seems like cell has some threshold for this protein levels. Did authors find that there is a threshold for overexpression of this protein?
3. The ngFBP2 seems to be a better strain even though the mRNA levels of FBPase were significantly lower than NgFBP1. Is there any explanation that authors would like to give for this?
4. In my opinion, use of simulated values for lipids is a good indication but not a replacement of lab experiments. Since authors were not able to show experimentally higher lipid yields I would suggest to provide experimental data to support this claim.
5. What are authors thoughts on using this engineered strain? Increase in biomass isn't always advantageous and depends a lot on type of product. What are future strategies?
Comments on the Quality of English LanguageNA
Author Response
Dear reviewer,
Thank you for reviewing our manuscript ID: ijms-2947818, which entitled " Metabolomics reveals the impact of overexpression of cytosolic fructose-1,6-bisphosphatase on Photosynthesis and Growth in Nannochloropsis gaditana", and providing additional comments and recommendations, which have not only made this work scientifically more accurate, but also helped to improve its readability to a broad audience. Our revised manuscript has been greatly improved according to the comments. All questions and suggestions were addressed and corrected accordingly. Please see the attachment.
Thank you and best regards.
Sincerely,
Hantao Zhou
State Key Laboratory of Marine Environmental Science, Xiamen University
College of Ocean and Earth Sciences, Xiamen University
Xiamen, Fujian 361000, China
E-mail: htzhou@xmu.edu.cn

Reviewer 2 Report
Comments and Suggestions for AuthorsТhe submitted manuscript “Metabolomics reveals the impact of overexpression of cytosolic fructose-1,6-bisphosphatase on Photosynthesis and Growth in Nannochloropsis gaditana” by authors Zhang et al represents an interesting investigation on the ability of microalgae Nannochloropsis gaditana, transformed to over express the cytosolic enzyme fructose-1,6-bisphos-phatase (cyFBPase), to increase the biomass of the culture and the amount of C20 unsaturated fatty acids and metabolites as glycine, protoporphyrin, thioglucosides, pantothenic acid, CoA, and glycerophospholipids. The findings could be of interest for readers interested in original biotechnology approach for obtaining higher biomass from micro algae.
The manuscript is well structured and well written.
I have the following recommendations:
- Line 118 and 122 – the first sentences are not completed. Probably the authors intended to give and additional subsections.
- Through the whole Introduction section CO2 – 2 to be in subscript.
- The letters in Fig. 1 are very small and difficult to read.
- Fig. 4 – In the figure legend in stated that the results about “chlorophyll content” are displayed, while in the figure are given Fv/Fm and rETR. In addition, while the values for Fv/Fm of WT are increasing with time, the values for effectivity of PSII, for WT and for both transformed cultures, are lower for the 6th day in comparison with 3rd day.
- Line 296-299 – It is stated that the “chlorophyll content” is “assessed by chlorophyll fluorescence”. It is not described in the Materials and Methods section.
- Line 371-372 – It is stated that the “overexpression of the cyFBPase had no effect on the total lipid content of the cells”. At the same time results in Fig. 6, B indicate that “lipid yield” and C – the content of C14:0, C18:1, C18:2, C20:4 and C20:5 is increased in transformed strains. Decline was detected only for C16:0 and C16:1. Does it means that the observed increase in the mentioned species is compensated by the decline in the 2 that are decreased? Then, the results in panel B indicating that lipid yield in transformed microalgae on the 4th and 12th day are not correct.
- In subsection 4.3. in Materials and Methods – “Chlorophyll content and chlorophyll fluorescence data were analyzed using two-way ANOVA” it is not indicated how the data about chlorophyll content was estimated by PAM in order to be statistically analyzed.
- Line 555-556, 557-559 – no equations are given. The formulas about Fv/Fm and effective quantum yield of PSII (Y(II)) are also missing as well as references for calculation.
- In section Materials and Methods is missing the subsection Statistics.
- English language style and gramma are acceptable.
The manuscript can be accepted after major revision.
Comments on the Quality of English LanguageEnglish language style and gramma are acceptable.
Author Response

(The authors gave the same response as above.)

Round 2
Reviewer 2 Report
Comments and Suggestions for Authors
The manuscript was significantly improved, taking into account the recommendations.
I have no recomendations for the English language style.
Can be accepted for publication